

# Multi-decadal changes in structural complexity following mass coral mortality on a Caribbean reef

George Roff[1], Jennifer Joseph[2], Peter J Mumby[1]

[1] Marine Spatial Ecology Lab, School of Biological Sciences, University of Queensland, QLD 4072 Australia

[2] College of Earth, Ocean & Environment, University of Delaware, Newark, DE, USA

*Correspondence to*: George Roff (g.roff@uq.edu.au)

**Abstract.** In recent decades, extensive mortality of reef-building corals throughout the Caribbean region has led to erosion of reef frameworks and declines in biodiversity. Using field observations, structural models and high-precision U-th dating methods, we quantify changes in structural complexity in the major framework building coral *Orbicella annularis* over a 20-year period at Long Cay (Belize). Despite extensive mortality following the mass coral bleaching event of 1998, structural complexity of frameworks remained largely unchanged between 1998 (rugosity index = $2.35 \pm 0.1$) and 2018 ($2.29 \pm 0.1$). Reef-scale structural complexity was maintained through rapid growth of surviving ramets ($0.69 \pm 0.1$ cm yr$^{-1}$) offsetting slower bioerosion of dead ramets ($-0.11 \pm 0.16$ cm yr$^{-1}$). Despite apparent stability of structural complexity at reef-scales, bioerosion of individual dead ramets over two decades led to declines in microhabitat complexity, with an overall reduction of the depth of microhabitats within frameworks. Altered microhabitat complexity appears to have negative effects for cryptic fauna, with the grazing urchin *Echinometra viridis* declining from $1.5 \pm 0.4$ per m$^2$ in 1998 to $0.02 \pm 0.02$ per m$^2$ in 2018. Changes in microhabitat complexity have the potential to alter ecological interactions that can impact recovery dynamics on coral reefs in ways that are undetectable using reef-scale metrics of structural complexity.

## 1. Introduction

The massive coral *Orbicella annularis* (Ellis and Solander, 1786) is the most abundant coral on modern-day Caribbean coral reefs (Dawson, 2006). As an ecosystem engineer, *O.annularis* plays a critical ecosystem role as a framework building coral (Geister, 1977) providing reef-scale structural complexity that supports a diverse range of fish (Alvarez-Filip et al., 2011) and invertebrate (Idjadi and Edmunds, 2006) assemblages. From a geological perspective, *O. annularis* first appeared in the late Pliocene to early Pleistocene (~2 - 1.5 million years ago, (Budd and Klaus, 2001)) and linear extension of ~10 mm yr$^{-1}$ resulted in the accretion of modern day Caribbean coral reefs at a rate of ~3.3 m kyr$^{-1}$ (Gischler, 2008). Accretion of reef frameworks is a key ecosystem function, providing critical habitat for reef organisms and physical coastline protection from wave erosion (Moberg and Folke, 1999).



While Caribbean coral reefs have exhibited remarkable ecological persistence over geological timeframes (Pandolfi and Jackson, 2006), reefs throughout the region have declined over the past century due to climate change, overfishing and poor water quality (Knowlton and Jackson, 2008). In recent decades, populations of *O. annularis* have been decimated by recurrent coral bleaching events (Szmant and Gassman, 1990), disease outbreaks (Pantos et al., 2003;Bruno et al., 2003), and increasing negative interactions with macroalgae (Ferrari et al., 2012) resulting in region-wide declines of *O. annularis* and shifts in

population structure to higher densities of smaller colonies (Nugues, 2002;Bruckner and Bruckner, 2006;Edmunds, 2015). Following the loss of living coral tissue, the exposed coral skeleton is subjected to a gradual process of bioerosion – the biologically induced weakening and breaking down of coral skeletons (Glynn and Manzello, 2015).

On modern-day coral reefs, grazing parrotfish are the primary bioeroders, contributing between 79 - 84% of total bioerosion (Perry et al., 2014). Parrotfish scrape and excavate the external surfaces of coral reef frameworks (Clements et al., 2017),

producing large volumes of carbonate sediments. Colonisation of exposed framework substrates by a broad range of micro and macro-boring organisms, including sponges, endolithic algae, fungi, and boring invertebrates further weakens coral skeletons, making them more susceptible to physical erosion (Hutchings, 1986). The erosion of reef frameworks in the Caribbean over the past decades has resulted in a loss of reef-scale structural complexity, a process described as "flattening" (Alvarez-Filip et al., 2009). This region-wide loss of structural complexity has led to substantial declines in ecosystem

functioning, including reductions in biodiversity (Newman et al., 2015) and losses of fisheries productivity (Rogers et al., 2018).

As the processes of accretion and erosion of coral reef frameworks occur over decadal to centennial time frames (Glynn and Manzello, 2015), few long-term in-situ studies of bioerosion exist (but see Kuffner et al., 2019). Here, we return to a multi-decadal study of Caribbean coral frameworks in Glovers Reef, Belize (Roff et al., 2015;Mumby, 1999). In 1998, extreme

thermal stress and the impact of hurricane Mitch resulted in extensive coral mortality throughout the Mesoamerican Barrier Reef (Mumby, 1999;Aronson et al., 2002;Mumby et al., 2005). Projections of future thermal stress events indicate that such mass bleaching events may become an annual occurrence by 2040 (van Hooidonk et al., 2015). These events are predicted to have a strong negative impact on carbonate budgets and reef accretion (Kennedy et al., 2013;Perry et al., 2013), yet the landscape-scale impacts on the structure and function of coral frameworks are not well understood. Here we quantify

multidecadal changes in *O. annularis* frameworks at Long Cay (Glovers Reef) and quantify long-term changes in structural complexity and microhabitat complexity.

## 2. Results and discussion

Prior to the 1998 mass bleaching event, frameworks at Long Cay (Glovers Reef, Figure 1a) were dominated by large long-lived *O. annularis* colonies (Figure 1b). Surveys at Long Cay in June 1998 (10 m depth) revealed high levels of healthy *O.*

*annularis* cover (68 ± 14%). Major coral bleaching occurred in September 1998 following a period of calm weather and



elevated water temperatures, with 70 to 90% of *O. annularis* colonies exhibiting full or partial bleaching (Roff et al., 2015). Extensive wave damage from Hurricane Mitch in November 1998 compounded the impacts of bleaching, with ~85% of *O. annularis* colonies exhibiting partial mortality (Roff et al., 2015). By June 2000, cover of *O. annularis* had declined to $20 \pm 5$ %, and remained at ~15% in the decades following the mass bleaching event despite repeat minor hurricane disturbance (Figure

1c). Genetic analysis of these surviving *O. annularis* colonies at Long Caye indicates high levels of genotypic diversity compared to other reefs around the Caribbean (Foster et al., 2013). Field surveys in 2018 revealed high levels of macroalgal cover and extensive partial mortality within *O. annularis* colonies (Figure 2a), with low survival of *O. annularis* ramets within colonies (averaging $14.1 \pm 11$ % SD).

To determine changes in *O. annularis* frameworks at different scales, we calculated two metrics of habitat complexity: i)

microhabitat complexity at the scale of individual ramets (centimetres), and ii) structural complexity at the scale of whole colonies (metres). Microhabitat complexity of *O. annularis* provides critical permanent refugia for small reef-fish and cryptic invertebrates among ramets (Almany, 2004;Sammarco, 1982), and structural complexity at colony scales can reduce the foraging efficiency of predators and limit prey detection (Beukers and Jones, 1998).

## 2.1 Multi-decadal changes in microhabitat complexity

Prior to the 1998 mortality event, ramet heights within colonies were consistent, resulting in an even hemispherical colony appearance (Figure 1b). Surveys in 2018 revealed that surviving colonies exhibited a characteristic "serrated" topography, in that the isolated surviving ramets protruded above the remaining dead colony (Figure 2a, Figure S1). We hypothesised that differences in growth among surviving ramets and the erosion of dead ramets would result in changes to microhabitat complexity. To test this hypothesis, we measured height differences between live and dead ramet pairings from *O. annularis*

colonies at Long Cay in successive decades, 2007 and 2018 (Figure 2b). Measurements of ramet pairings in 2007 indicated a significant difference in height between "live" and "dead" ramets of $3.1 \pm 2.9$ cm ($p < 0.01$, Figure 2c), a trend that had increased to $15.5 \pm 3.8$ cm in 2018 ($p < 0.01$, Figure 2c). No significant difference in height was observed between "live-live" or "dead-dead" ramet pairings in either 2007 or 2018 ($p > 0.05$), implying that processes of growth and/or erosion occur evenly among living and dead ramets (Figure S1).

Using high-precision U-th dating methods and CT scan reconstructions, we previously quantified annual rates of external bioerosion and growth (linear extension) of *O. annularis* colonies at Long Cay between 1998 and 2011 (Roff et al., 2015). Estimates from U-th dates indicated an average erosion rate of $-0.11 \pm 0.16$ cm yr$^{-1}$, which was in close agreement with model predictions of external erosion by excavating parrotfish (Roff et al., 2015). Within the same time frame, linear extension (growth) of *O. annularis* ramets (as quantified by density banding of skeletons) was more than sixfold higher ($0.69 \pm 0.1$ cm

yr$^{-1}$) than surface bioerosion (Roff et al., 2015). Based upon these parameters, an estimate of ramet height difference between 1998 and 2018 would indicate a ~ $13.8 \pm 2$ cm vertical growth of surviving *O. annularis* ramets, while bioerosion results in ~



-2.2 ± 1 cm loss of ramet height. The modelled net outcome of erosion and growth of ramets of Δ16 cm between 1998 – 2018 (Figure 2d) is in close agreement to the observed differences between live and dead ramets in 2018 of 15.4 ± 1.1 cm, indicating that the six-fold higher growth of surviving ramets results in the observed "serrated" pattern of *O. annularis* colonies (Figure 95    2b).

## 2.2 Multi-decadal changes in structural complexity

To determine changes in structural complexity at colony scales, we created a structural model of *O. annularis* colonies (Fig 3a) parameterised using field data collected at Long Cay (see methods for full details and model code). Briefly, the model simulates annual changes in structural complexity of 1000 *O. annularis* colonies between 1998 – 2018 through growth of 100    surviving ramets and erosion of dead ramets (Figure 3a). Structural complexity within each simulated colony was measured using the rugosity index (*R*, Figure 3), a common index of rugosity on coral reefs where a flat surface has an *R* of 1, and larger numbers reflect a greater degree of structural complexity (Alvarez-Filip et al., 2009). Prior to the bleaching event in 1998, uniform growth of living *O. annularis* ramets resulted in *R* values of 2.36 (95% CI: 2.3 - 2.4). Following the bleaching event, extensive partial mortality resulted in erosion of dead ramets and growth of surviving ramets. Hindcasting long-term changes 105    in structural complexity between 1998-2018 revealed high variability among colony trajectories, with 58% exhibiting declining rugosity, 41% exhibiting increases in rugosity, and 1% exhibiting no change. Despite such variance, overall structural complexity was remarkably stable (Fig 3b), and average values of *R* in 2018 were within the range of pre-mortality levels (*R* = 2.31, 95% CI: 2.28 - 2.34). Sensitivity analysis of the model indicates that changes in structural complexity between 1998 and 2018 (Δ*R*) was weakly correlated to colony size (ρ = 0.1, 95% CI: 0.04 – 0.15) in that smaller colonies experienced declines 110    in structural complexity (Figure 3c). Post-disturbance survival was a key driver of change in rugosity (ρ = 0.79, 95% CI: 0.75 – 0.8), in that colonies that experienced high levels of within colony mortality (>80%) experienced declines in structural complexity (Figure 3d), whereas colonies that experienced higher survival rates (>20% surviving ramets within colonies) exhibited increases in structural complexity (Figure 3d). Comparisons of structural complexity reconstructed from paired photographs of *O. annularis* colonies at Long Cay in 2003 and 2018 (Figure 4a,b) validate model predictions, and support 115    wider field observations at Glovers Reef in 2018 of surviving ramets protrude above dead *O. annularis* frameworks (Figure S1).

Losses of structural complexity following disturbance are primarily thought to be driven by processes of erosion (Alvarez-Filip et al., 2009;Sheppard et al., 2002;Glynn, 1988), in that physical disturbance and chemical dissolution, combined with 120    intense internal and external bioerosion flattens structurally complex coral reef structure. If processes of bioconstruction from coral growth are weak, reefs remain in a degraded and flattened state, as has occurred in several areas of the Caribbean to varying extents (Alvarez-Filip et al., 2009).  Here, we document changes in microhabitat complexity following mass coral mortality that appear to be driven primarily by growth of surviving ramets of *O. annularis* rather than through processes of bioerosion. High levels of genotypic diversity in *O. annularis* at Long Caye (Foster et al., 2013) and population connectivity





to other reefs throughout the western Caribbean (Foster et al., 2012) implies that Long Caye is not unique, and differential
      growth of surviving ramets may lead to changes in in structural complexity elsewhere in the Caribbean where growth rates
      exceed erosion. At colony scales, changes in microhabitat complexity do not appear to have translated into changes in reef
      complexity, as the erosion of dead ramets is offset by growth of surviving ramets. In colonies of *O. annularis* that experienced
      partial mortality, the wide spacing among surviving ramets (Figure 2c) allows access to previously protected skeleton by

grazing parrotfish (see initial phase *Sparisoma viride*, Figure 1c). Such increased access to parrotfish appears to have
      accelerated erosion rates on the sides of surviving ramets, resulting in a narrowing of ramets (Figure S1) which increases
      susceptibility to physical breakage (Hein and Risk, 1975).

      Long-term records of bioerosion over ecologically meaningful timescales are rare, yet a recent study (Kuffner et al., 2019)
      reporting exceptionally rapid rates of erosion of reef frameworks (maximum 1.63 cm yr$^{-1}$) in the Florida Keys provides

important insight into heterogeneity of framework erosion throughout the wider Caribbean. The low rates of *O. annularis*
      bioerosion at Long Cay reported in the present study (-2.2 ± 1 cm over 20 years) compared to the Florida Keys (9.4 ± 5.6 cm
      over 17 years) is likely due to differences in skeletal density (1.9 g cm$^3$, (Roff et al., 2015) vs 1.12 g cm$^3$, (Halley et al., 1994))
      driven by the highly productive windward location of Glovers Reef, but may also reflect regional differences in the structure
      of bioeroding herbivore assemblages. These two studies likely reflect the extremes of the bioerosionary spectrum on Caribbean

reefs, and future research should focus on understanding factors that render some frameworks more resilient than others. While
      secondary cementation played an important role in hardening Holocene reef frameworks (Gischler and Hudson, 2004), the
      geochemical evidence (consistent initial uranium concentrations) within modern *O. annularis* skeletons from Long Cay (Roff
      et al., 2015) suggests that secondary cementation may not necessarily play an important reinforcing role in modern corals over
      at least decadal time frames.

**2.3 Changing functional roles of structural complexity**

      At reef scales, *O. annularis* forms structurally complex frameworks that underpin species richness on Caribbean coral reefs
      (Newman et al., 2015). Prior to the 1998 disturbance events, the narrow crevices between *O. annularis* ramets provided critical
      refuge for scleractinian coral recruits (Mumby, 1999), juvenile and small-bodied Caribbean reef fish (Nemeth, 1998;Alvarez-
      Filip et al., 2011), and the eroding echinoids *Echinometra viridis* ((Sammarco, 1982), Figure 5a) and juvenile *Diadema*

*antillarum* (Lessios, 1998). *E.viridis* are highly abundant in patch reefs and lagoonal reefs throughout Belize (Brown-Saracino
      et al., 2007), yet are historically less common on deeper, exposed fore-reef habitats such as Long Cay (PJM pers. obs.). *D.
      antillarum* have been historically rare on the windward reef slope at Glovers Reef (PJM pers. obs.) following the Caribbean-
      wide die-off in the early 1980's (Lessios, 1988), surveys prior to the mass bleaching event in 1998 indicate that *O. annularis*
      frameworks supported a population of the smaller urchin *E. viridis* at densities of 1.1 ± 0.6 individuals per m$^2$ (Figure 5b). As

*E. viridis* are largely limited to crevice microhabitats due to high rates of predation (McClanahan, 1999), we hypothesise that
      the observed long-term changes in microhabitat complexity among crevices at Long Cay allows for increased access for



predatory fish such as triggerfish (Balistidae) and porgies (Sparidae). In 2007, *E. vidiris* was observed among *O. annularis* microhabitats at a comparable density to 1998 surveys, averaging $1.5 \pm 0.4$ individuals per m$^2$ to a maximum of 14.3 individuals per m$^2$ (Figure 4b). Ten years later, in 2018, we repeated surveys across the same study area at Long Cay, and only a single *E.*

*viridis* was recorded in 40 colonies, resulting in an average density of just $0.02 \pm 0.02$ individuals per m$^2$ (Figure 5b). While further experimental work is needed to quantify size thresholds of refugia by which urchins escape predation, these observations are consistent with the hypothesis that bioerosion of reef frameworks results in reduced crevice depth, which in turn affects *E. viridis* densities by allowing for increased access for invertivorous fish, resulting in higher urchin mortality. Declines in the minimum ramet depth from $6.6 \pm 3.9$ cm in 2007 to $4.8 \pm 2.1$ cm (Figure 2c) support this hypothesis, and are

consistent with our previous U-th estimates of bioerosion at Long Cay (Roff et al., 2015). As parrotfish preferentially target convex surfaces of dead coral substrates (Roff et al., 2011), bioerosion of ramet edges may also have increased the aperture of crevices, further facilitating access to invertivores and diminishing refuge potential. Population dynamics of urchins are complex, and result from complex interactions between top-down and bottom-up factors (Tebbett and Bellwood, 2018). Higher biomass of invertivores inside of marine protected areas can substantially increase predation pressure on urchins (Harborne et

al., 2009), and may explain the rapid decline in *E. viridis* at Long Caye following diminished refuge potential between surveys.

Densities of *E. viridis* at the deeper exposed fore-reef habitat at Long Cay are substantially lower than reported for other shallow patch reef and lagoonal reef habitats in Belize (as high as $40 \pm 7$ individuals per m$^2$, (Brown-Saracino et al., 2007)). Yet even at low densities (< 2 individuals per m$^2$) *E. viridis* density is positively related to coral cover (Bologna et al., 2012), implying a functional link. As *E. viridis* can play an important role in structuring reef communities by maintaining algal free

space within ramet crevices (Figure 5a), in turn facilitating coral recruitment (Sammarco, 1982), losses of these urchins may have resulted in small-scale increases in macroalgal cover within *O. annularis* framework microhabitats – notably *Lobophora* spp. (Figure 1c) – which can reduce coral recruitment and impede future recovery potential (Mumby et al., 2007). As an ecosystem engineer, the structural complexity constructed by *O. annularis* provides critical refuge for a diverse range of invertebrate fauna (Buss and Jackson, 1979;Idjadi and Edmunds, 2006). As diversity is positively related to structural

complexity and not coral cover (Idjadi and Edmunds, 2006), we expect to see similar declines in other motile and sessile invertebrate taxa that seek refuge in *O. annularis* microhabitats.

### 3. Structural complexity and decline of reef frameworks in the 21st century

In recent decades, declines in coral cover and losses of keystone species have resulted in region-wide reductions of structural complexity throughout the Caribbean (Alvarez-Filip et al., 2009). Here we highlight how small-scale changes in microhabitat

complexity have the potential to alter ecological interactions that can impact recovery dynamics in ways that are undetectable using standardised metrics of structural complexity. While losses of key microhabitat complexity may have cascading effects on diversity and ecosystem function, our results indicate that structural complexity at reef scales can be remarkably robust. Given the recent widespread recruitment failure of Caribbean corals (Hughes and Tanner, 2000) and low recruitment rates of



*O. annularis* in general (Edmunds, 2002), the potential for recovery and long-term future of *O. annularis* frameworks at Long

Cay in the 21st century is unclear. However, two decades after mass mortality at Long Cay, levels of coral cover in 2018 remain above the threshold of live coral cover of ~10% needed to maintain a positive state of reef accretion (Roff et al., 2015;Perry et al., 2013). Ongoing bioerosion from micro and macroborers (Roff et al., 2015) will likely weaken skeletal structural integrity in *O. annularis* (Highsmith et al., 1983), facilitating mechanical breakage and storm-driven loss of now protruding surviving ramets (Figure S1), likely resulting in non-linear increases of framework loss over longer timescales (2050 and beyond).

# 4. Methods

The study was conducted in Long Cay (Glovers Reef, Belize, Figure 1a). The reef framework at Long Cay is formed primarily from colonies of *Orbicella annularis* (Ellis and Solander, 1786), which experienced widespread mortality following anomalously high water temperatures (29–32 °C) between early September and mid-November 1998 and hurricane Mitch which occurred simultaneously (Mumby, 1999). Field data were collected in 1998, 2007 and 2018 from an area of *O. annularis*

dominated framework of approximately 400 m$^2$ at a depth of 6-12m.

## 4.1 Microhabitat complexity

Surveys and measurements of *O. annularis* colonies were conducted at 6-12 m depth at Long Cay in March 2007 and May 2018. At both time points, colonies of *O. annularis* were selected among the framework at random using a system of fin-kicks and compass bearings. A distance of ~5m was maintained between measured colonies, and ramets with signs of recent death

were avoided. For each colony, we selected neighbouring ramet pairs from the central part of the colony ($n$ = 2-3 per colony) and assigned it to one of three states: either "live-live" pairing, "live-dead" pairing or "dead-dead" pairing ($n$ = 20 ramet pair measurements for each state in 2007, $n$ = 30 in 2018). The difference in height between each neighbouring ramet pair was quantified using a ruler or calipers to determine differences in the microhabitat complexity within *O. annularis* colonies. Differences between 2007 and 2018 were tested with a linear mixed effects model in R software (R Development Core Team,

2019). with "year" and "state" as fixed factors.

## 4.2 Structural complexity

To assess changes in rugosity at a colony scale in the two decades following the mass mortality, we created a structural model of *O. annularis* colonies parameterised using field data collected at Long Cay (see Supplementary code). *O. annularis* colonies were modelled using a simple cross-sectional topography of ramets (Figure 4a). Colony widths were determined from in-situ

measurements of 95 colonies at Long Cay in 2000, and ramet heights and widths measured from 30 ramets within colonies in 2000 (Roff et al., 2015). The model simulates 1000 colonies of *O. annularis* randomly sampling from colony width measurements and uses measurements of ramet diameter and ramet spacing to determine the number of ramets within colonies. Prior to the 1998 mortality event, the ratio of live to dead ramets within colonies was determined from surveys of pre-disturbance *O. annularis* in 1998, where 97.8% of *O. annularis* ramets were alive (Roff et al., 2015). The ratio of live to dead

ramets twenty years after the 1998 mortality event was determined from surveys of 25 colonies of *O. annularis* at Long Cay in 2018. Surviving ramets were subject to annual linear extension (sampled at random from CT scan derived skeletal growth measurements at Long Cay between 2006-2011, (Roff et al., 2015)), while dead ramets underwent annual external linear bioerosion (sampled at random from U-th derived measurements of *O. annularis* at Long Cay between 1998-2011, (Roff et al., 2015)). Annual changes in structural complexity within each simulated colony between 1998-2018 was measured using

the rugosity index (*R*, (Alvarez-Filip et al., 2009)), a ratio between the width of the colony and the external surface (i.e. sum of ramet heights, ramet width and inter-ramet spacing). Correlations between changes in rugosity (1998-2018) and colony width, post-bleaching survival, and both number of live and dead ramets within simulated colonies was determined using Pearson's product moment correlation coefficient ($\rho$) with the cor.test function ('stats' package) in R software (R Development Core Team, 2019).

**4.3 Urchin densities**

  To examine local ecological impacts of changes in ramet size distribution, we censused the density of the urchin *Echinometra viridis,* which occupies the interstitial space between ramets. Surveys of at Long Cay in 1998 were conducted using five 10×0.5 m transects, and found that urchins were dominated by *E.viridis* within ramets of *O. annularis* framework (Mumby et al., 2005). In subsequent survey years (2007 and 2018), urchin surveys were conducted on a per colony basis, and the number

of urchins within each O. annularis colony was standardised to the colony area to give individuals per m$^2$ (*n* = 50 colonies, 2018: *n* = 40 colonies).

**Data availability**

Complete R code for structural model is provided in the supplementary, and underlying survey data will be made available on Dryad Digital Repository.

**Supplement**

Figure S1 Characteristic "serrated" topography of *O. annularis* colonies at Long Caye in 2018 with isolated surviving ramets protruded above the remaining dead colony.

**Author contributions**

Survey data: PJM, GR, JJ; structural complexity model: GR; writing – original draft: GR; writing – review
and editing: GR, PJM





**Competing interests.**

The authors declare that they have no conflict of interest.

**Acknowledgements**

We thank the Belize Department of Fisheries for permission to undertake this study and support from the Smithsonian research
station at Carrie Bow Caye. This study was funded by the ARC Centre of Excellence for Coral Reef Science. This is the XXX
research contribution from the Carrie Bow Caye Marine Station.

**Refereneces**

Almany, G. R.: Does increased habitat complexity reduce predation and competition in coral reef fish assemblages?, Oikos, 106, 275-284,
255    2004.
Alvarez-Filip, L., Dulvy, N. K., Gill, J. A., Cote, I. M., and Watkinson, A. R.: Flattening of Caribbean coral reefs: region-wide declines in architectural complexity, Proceedings of the Royal Society B-Biological Sciences, 276, 3019-3025, 2009.
Alvarez-Filip, L., Gill, J. A., and Dulvy, N. K.: Complex reef architecture supports more small-bodied fishes and longer food chains on Caribbean reefs, Ecosphere, 2, Unsp 118
10.1890/Es11-00185.1, 2011.
Aronson, R. B., Precht, W. F., Toscano, M. A., and Koltes, K. H.: The 1998 bleaching event and its aftermath on a coral reef in Belize, Marine Biology, 141, 435-447, 10.1007/s00227-002-0842-5, 2002.
Beukers, J. S., and Jones, G. P.: Habitat complexity modifies the impact of piscivores on a coral reef fish population, Oecologia, 114, 50-59, 1998.
Bologna, P. A. X., Webb-Wilson, L., Connelly, P., and Saunders, J. E.: A new baseline for *Diadema antillarum*, *Echinometra viridis*, *E. lucunter*, and *Eucidaris tribuloides* populations within the Cayos Cochinos MPA, Honduras, Gulf and Caribbean Research, 24, 1-5, 2012.
Brown-Saracino, J., Peckol, P., Curran, H. A., and Robbart, M. L.: Spatial variation in sea urchins, fish predators, and bioerosion rates on coral reefs of Belize, Coral Reefs, 26, 71-78, 10.1007/s00338-006-0159-9, 2007.
Bruckner, A. W., and Bruckner, R. J.: Consequences of yellow band disease (YBD) on Montastraea annularis (species complex) populations
on remote reefs off Mona Island, Puerto Rico, Diseases of Aquatic Organisms, 69, 67-73, 2006.
Bruno, J. F., Petes, L. E., Harvell, C. D., and Hettinger, A.: Nutrient enrichment can increase the severity of coral diseases, Ecology Letters, 6, 1056-1061, 10.1046/j.1461-0248.2003.00544.x, 2003.
Budd, A. F., and Klaus, J. S.: The origin and early evolution of the Montastraea "annularis" species complex (Anthozoa : Scleractinia), Journal of Paleontology, 75, 527-545, 2001.
Buss, L. W., and Jackson, J. B. C.: Competitive networks: non-transitive competitive relationships in cryptic coral reef environments., American Naturalist, 113, 223-234, 1979.
Clements, K. D., German, D. P., Piche, J., Tribollet, A., and Choat, J. H.: Integrating ecological roles and trophic diversification on coral reefs: multiple lines of evidence identify parrotfishes as microphages, Biological Journal of the Linnean Society, 120, 729-751, 2017.
Dawson, J. P.: Quantifying the colony shape of the Montastraea annularis species complex, Coral Reefs, 25, 383-389, 10.1007/s00338-006-
280    0124-7, 2006.
Edmunds, P. J.: Long-term dynamics of coral reefs in St. John, US Virgin Islands, Coral Reefs, 21, 357-367, 10.1007/s00338-002-0258-1, 2002.
Edmunds, P. J.: A quarter-century demographic analysis of the Caribbean coral, Orbicella annularis, and projections of population size over the next century, Limnology and Oceanography, 60, 840-855, 10.1002/lno.10075, 2015.
Ellis, J., and Solander, D.: The Natural History of many curious and uncommon Zoophytes, collected from various parts of the Globe. Systematically arranged and described by the late Daniel Solander, Benjamin White & Son, London, 1786.





Ferrari, R., Gonzalez, M., and Mumby, P. J.: Size matters in the competition between corals and macroalgae., Marine Ecology Progress Series, 467, 77-88, 2012.

Foster, N. L., Paris, C. B., Kool, J. T., Baums, I. B., Stevens, J. R., Sanchez, J. A., Bastidas, C., Agudelo, C., Bush, P., Day, O., Ferrari, R.,
Gonzalez, P., Gore, S., Guppy, R., McCartney, M. A., McCoy, C., Mendes, J., Srinivasan, A., Steiner, S., Vermeij, M. J. A., Weil, E., and Mumby, P. J.: Connectivity of Caribbean coral populations: complementary insights from empirical and modelled gene flow, Molecular Ecology, 21, 1143-1157, 10.1111/j.1365-294X.2012.05455.x, 2012.

Foster, N. L., Baums, I. B., Sanchez, J. A., Paris, C. B., Chollett, I., Agudelo, C. L., Vermeij, M. J. A., and Mumby, P. J.: Hurricane-Driven Patterns of Clonality in an Ecosystem Engineer: The Caribbean Coral Montastraea annularis, Plos One, 8, ARTN e53283
10.1371/journal.pone.0053283, 2013.

Geister, J.: The influence of wave exposure on the ecological zonation of Caribbean coral reefs., Proceedings of the Third International Coral Reef Symposium, 1977, 23-29,

Gischler, E., and Hudson, J. H.: Holocene development of the Belize Barrier Reef, Sedimentary Geology, 164, 223-236, 10.1016/j.sedgeo.2003.10.006, 2004.

Gischler, E.: Accretion patterns in Holocene tropical coral reefs: do massive coral reefs in deeper water with slowly growing corals accrete faster than shallower branched coral reefs with rapidly growing corals?, International Journal of Earth Sciences, 97, 851-859, 10.1007/s00531-007-0201-3, 2008.

Glynn, P. W.: El Niño warming, coral mortality and reef framework destruction by echinoid bioerosion in the Eastern Pacific, Galaxea, 7, 129-160, 1988.

Glynn, P. W., and Manzello, D.: Bioerosion and coral reef growth: a dynamic balance, in: Coral Reefs in the Anthropocene, edited by: Birkeland, C., Springer, 67-97, 2015.

Halley, R. B., Swart, P. K., Dodge, R. E., and Hudson, J. H.: Decade-scale trend in sea water salinity revealed through δ18O analysis of *Montastraea Annularis* annual growth bands., Bulletin of Marine Science, 54, 670-678, 1994.

Harborne, A. R., Renaud, P. G., Tyler, E. H. M., and Mumby, P. J.: Reduced density of the herbivorous urchin Diadema antillarum inside a
Caribbean marine reserve linked to increased predation pressure by fishes, Coral Reefs, 28, 783-791, Doi 10.1007/S00338-009-0516-6, 2009.

Hein, F. J., and Risk, M. J.: Bio-erosion of coral heads - inner patch reefs, Florida Reef Tract, Bulletin of Marine Science, 25, 133-138, 1975.

Highsmith, R. C., Lueptow, R. L., and Schonberg, S. C.: Growth and bioerosion of three massive corals on the Belize barrier reef., Marine Ecology Progress Series, 13, 261-271, 1983.

Hughes, T. P., and Tanner, J. E.: Recruitment failure, life histories, and long-term decline of Caribbean corals, Ecology, 81, 2250-2263, 2000.

Hutchings, P. A.: Biological Destruction of Coral Reefs - a Review, Coral Reefs, 4, 239-252, Doi 10.1007/Bf00298083, 1986.

Idjadi, J. A., and Edmunds, P. J.: Scleractinian corals as facilitators for other invertebrates on a Caribbean reef, Marine Ecology-Progress Series, 319, 117-127, 2006.

Kennedy, E. V., Perry, C. T., Halloran, P. R., Iglesias-Prieto, R., Schonberg, C. H. L., Wisshak, M., Form, A. U., Carricart-Ganivet, J. P., Fine, M., Eakin, C. M., and Mumby, P. J.: Avoiding coral reef functional collapse requires local and global action, Current Biology, 23, 912–918, 2013.

Knowlton, N., and Jackson, J. B. C.: Shifting baselines, local impacts, and global change on coral reefs, Plos Biology, 6, 215-220, 10.1371/journal.pbio.0060054, 2008.

Kuffner, I. B., Toth, L. T., Hudson, J. H., Goodwin, W. B., Sathakopoulos, A., Bartlett, L. A., and Whitcher, E. M.: Improving estimates of coral reef construction and erosion with in situ measurements, Limnology and Oceanography, 10.1002/lno.11184, 2019.

Lessios, H. A.: Mass mortality of *Diadema antillarum* in the Caribbean: what have we learned?, Annual Review of Ecology and Systematics, 19, 371-393, 1988.

Lessios, H. A.: Shallow water echinoids of Cayos Cochinos, Honduras, Revista De Biologia Tropical, 46, 95-101, 1998.

McClanahan, T. R.: Predation and the control of the sea urchin Echinometra viridis and fleshy algae in the patch reefs of Glovers Reef, Belize, Ecosystems, 2, 511-523, 1999.

Moberg, F., and Folke, C.: Ecological goods and services of coral reef ecosystems, Ecological Economics, 29, 215-233, 1999.

Mumby, P. J.: Bleaching and hurricane disturbances to populations of coral recruits in Belize, Marine Ecology-Progress Series, 190, 27-35, 1999.

Mumby, P. J., Foster, N. L., and Fahy, E. A. G.: Patch dynamics of coral reef macroalgae under chronic and acute disturbance, Coral Reefs, 24, 681-692, 10.1007/s00338-005-0058-5, 2005.

Mumby, P. J., Harborne, A. R., Williams, J., Kappel, C. V., Brumbaugh, D. R., Micheli, F., Holmes, K. E., Dahlgren, C. P., Paris, C. B., and Blackwell, P. G.: Trophic cascade facilitates coral recruitment in a marine reserve, Proceedings of the National Academy of Sciences of the United States of America, 104, 8362-8367, 10.1073/pnas.0702602104, 2007.

Nemeth, R. S.: The effect of natural variation in substrate architecture on the survival of juvenile bicolor damselfish, Environmental Biology of Fishes, 53, 129-141, 1998.



Newman, S. P., Meesters, E. H., Dryden, C. S., Williams, S. M., Sanchez, C., Mumby, P. J., and Polunin, N. V. C.: Reef flattening effects on total richness and species responses in the Caribbean, Journal of Animal Ecology, 84, 1678-1689, 10.1111/1365-2656.12429, 2015.

Nugues, M. M.: Impact of a coral disease outbreak on coral communities in St. Lucia: What and how much has been lost?, Marine Ecology-Progress Series, 229, 61-71, 2002.

Pandolfi, J. M., and Jackson, J. B. C.: Ecological persistence interrupted in Caribbean coral reefs, Ecology Letters, 9, 818-826, Doi 10.1111/J.1461-0248.2006.00933.X, 2006.

Pantos, O., Cooney, R. P., Le Tissier, M. D. A., Barer, M. R., and Bythell, J.: The bacterial ecology of a plague-like disease affecting the Caribbean coral Montastrea annularis, Environmental Microbiology, 5, 370-382, 2003.

Perry, C., Murphy, G. N., Kench, P. S., Smithers, S. G., Edinger, E. N., Steneck, R. S., and Mumby, P. J.: Caribbean-wide decline in carbonate production threatens coral reef growth, Nature Communications, 4, 1402, doi:10.1038/ncomms2409, 2013.

Perry, C. T., Murphy, G. N., Kench, P. S., Edinger, E. N., Smithers, S. G., Steneck, R. S., and Mumby, P. J.: Changing dynamics of Caribbean reef carbonate budgets: emergence of reef bioeroders as critical controls on present and future reef growth potential, Proceedings of the Royal Society B-Biological Sciences, 281, 10.1098/rspb.2014.2018, 2014.

R Development Core Team: R: A language and environment for statistical computing. R Foundation for Statistical Computing, Vienna, Austria, 2019.

Roff, G., Ledlie, M. H., Ortiz, J. C., and Mumby, P. J.: Spatial Patterns of Parrotfish Corallivory in the Caribbean: The Importance of Coral Taxa, Density and Size, PLoS ONE, 6, e29133, 2011.

Roff, G., Zhao, J. X., and Mumby, P. J.: Decadal-scale rates of reef erosion following El Nino-related mass coral mortality, Global Change 360 Biology, 21, 4415-4424, 10.1111/gcb.13006, 2015.

Rogers, A., Blanchard, J. L., and Mumby, P. J.: Fisheries productivity under progressive coral reef degradation, Journal of Applied Ecology, 55, 1041-1049, 10.1111/1365-2664.13051, 2018.

Sammarco, P. W.: Echinoid grazing as a structuring force in coral communities - whole reef manipulations., Journal of Experimental Marine Biology and Ecology, 61, 31-55, 1982.

Sheppard, C. R. C., Spalding, M., Bradshaw, C., and Wilson, S.: Erosion vs. recovery of coral reefs after 1998 El nino: Chagos reefs, Indian Ocean, Ambio, 31, 40-48, 2002.

Szmant, A. M., and Gassman, N. J.: The effects of prolonged bleaching on the tissue biomass and reproduction of the reef coral *Montastraea annularis*, Coral Reefs, 8, 217-224, 1990.

Tebbett, S. B., and Bellwood, D. R.: Functional links on coral reefs: Urchins and triggerfishes, a cautionary tale, Marine Environmental 370 Research, 141, 255-263, 10.1016/j.marenvres.2018.09.011, 2018.

van Hooidonk, R., Maynard, J. A., Liu, Y. Y., and Lee, S. K.: Downscaled projections of Caribbean coral bleaching that can inform conservation planning, Global Change Biology, 21, 3389-3401, 10.1111/gcb.12901, 2015.




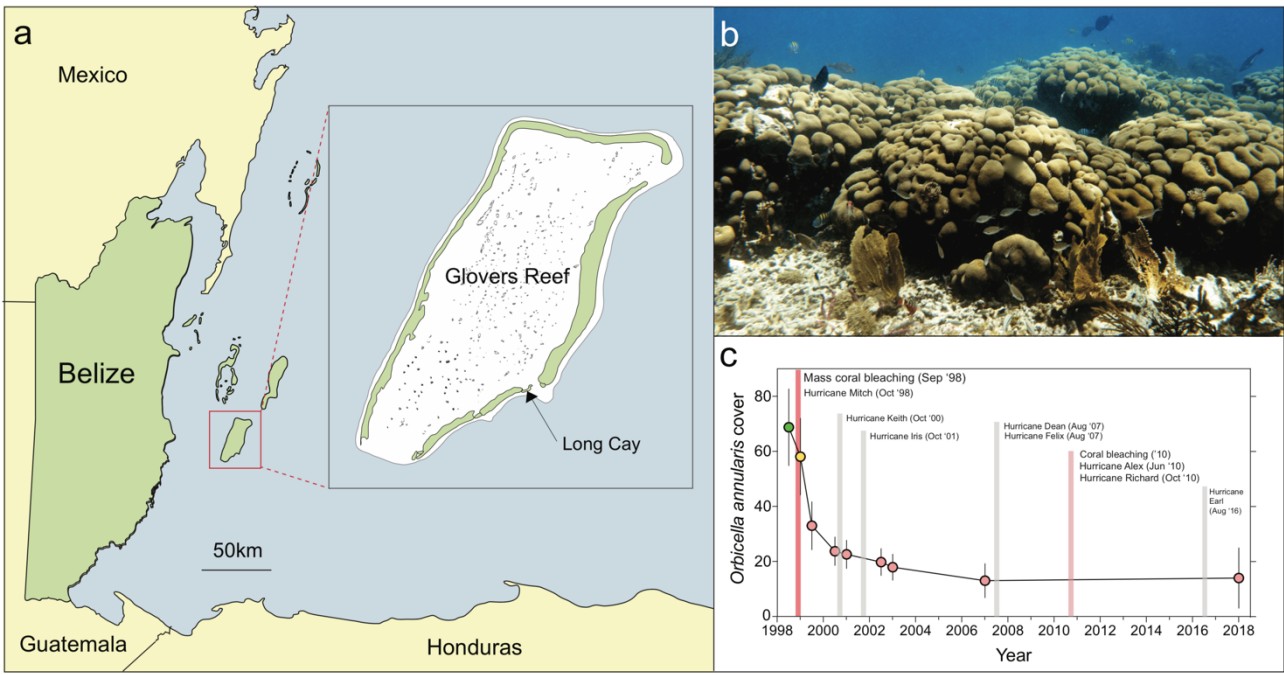

**Figure 1** a) Map of Belize and study site (inset) at Long Cay, Glovers Reef, b) Living *Orbicella annularis* colonies forming a structurally complex framework on a Caribbean reef, c) time series of *O. annularis* cover at Glovers Reef and major disturbance events between 1998 and 2018






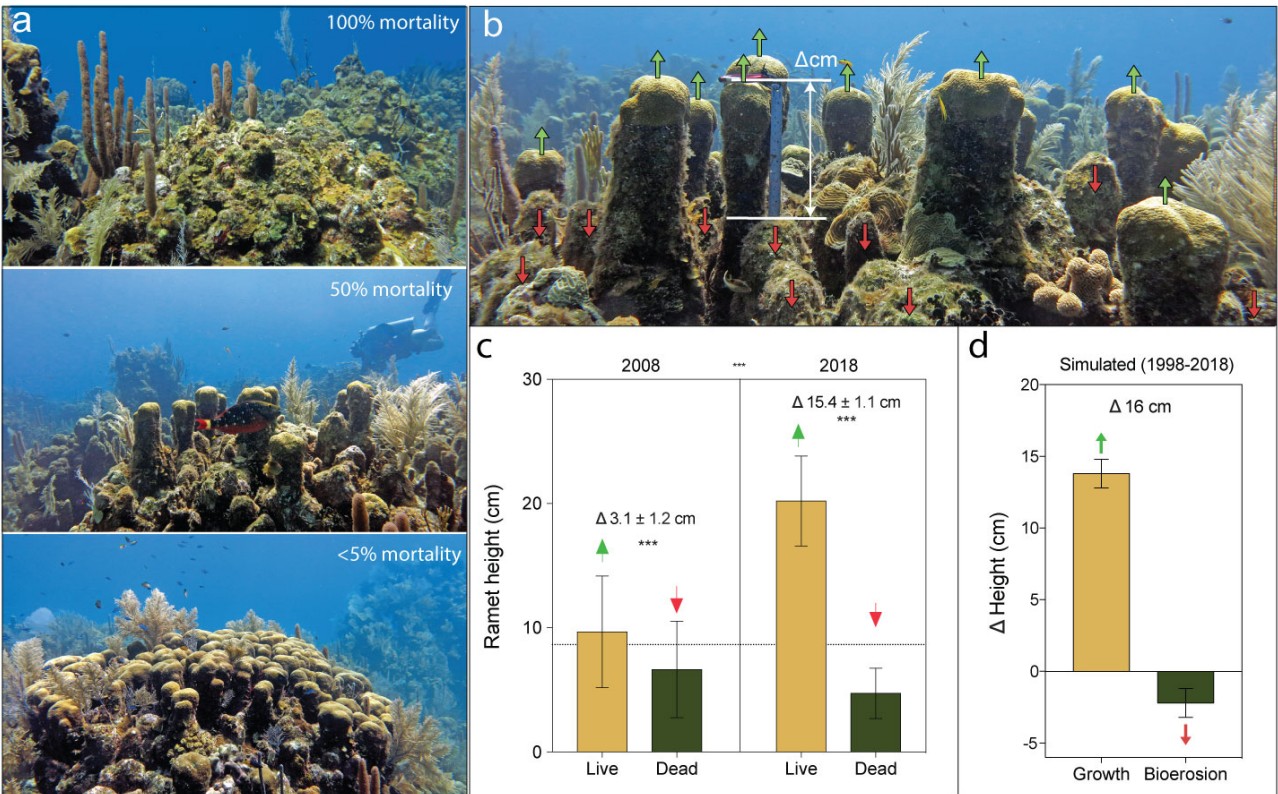

**Figure 2** a) *Orbicella annularis* colonies from Long Cay in 2018 indicating high survival following the mass bleaching event
(<5% mortality), complete (100%) mortality followed by algal overgrowth by *Lobophora variegata*, and partial mortality
(~50%) resulting in characteristic "serrated" pattern and gaps, b) *O. annularis* framework at Long Cay in 2007 showing growth
of surviving ramets (green arrows) protruding above the eroding dead ramets (red arrows) and difference in height between
live and dead ramets (scale ruler = 15cm), c) height of "live" and "dead" ramet pairings in 2007 and 2018, showing a significant
difference in height within years and a significant difference in average height of live ramets within years (*** = p < 0.001),
d) simulated growth and bioerosion between 1998 and 2018 of based upon a linear erosion rate of 0.11 ± 0.03 cm yr⁻¹ and
growth rate of 0.69 ± 0.1 cm yr⁻¹ (Roff et al., 2015).





**Figure 3** a) Cross-sectional structural model of *Orbicella annularis* indicating the method of estimating structural complexity
(*R* = surface perimeter / width), and changes in *R* in 1998 (100% live prior to mortality) and 2018 following mortality, b)
results of 1000 model simulations of changes in structural complexity between 1997 and 2018 (grey lines) and average values
across simulations (red line), c) change in structural complexity between 1998-2018 (Δ*R*) of 1000 simulated colonies against
colony size and d) proportion of colony survival / mortality.






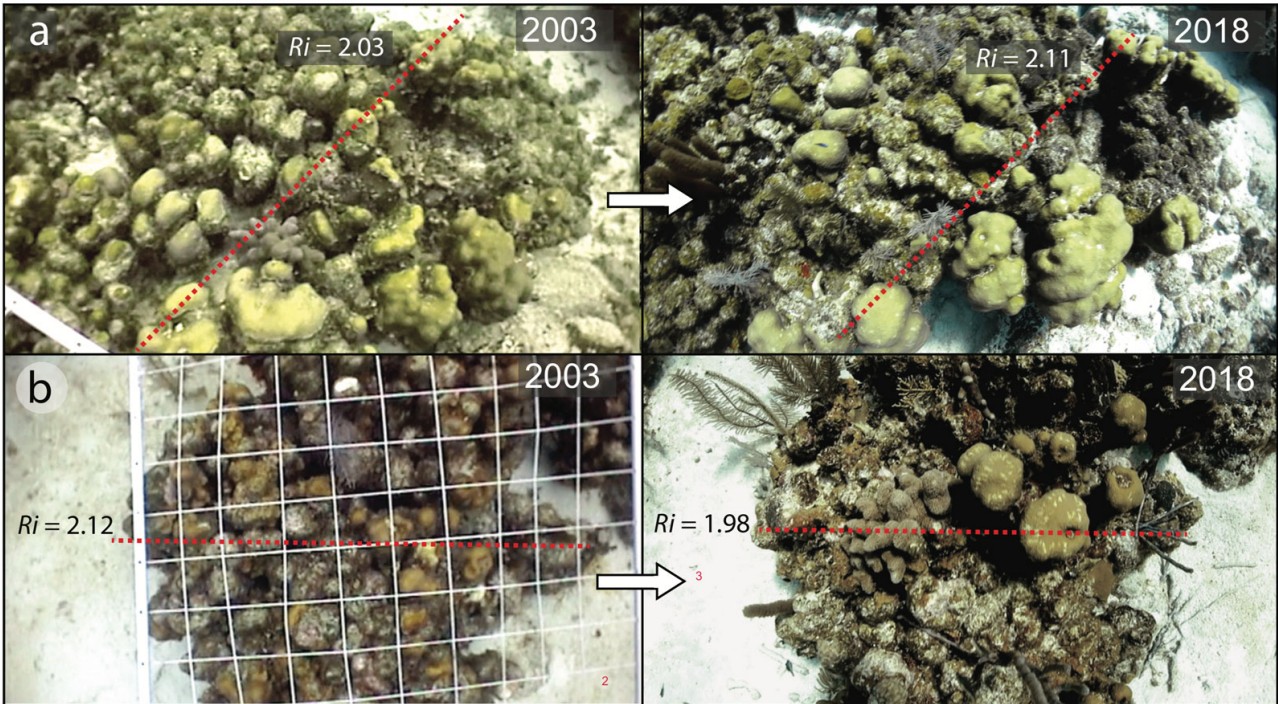

**Figure 4** a-b) paired photographs of *Orbicella. annularis* colonies from 2003 (5 years post-disturbance) and 2018 (20 years post-disturbance) and structural complexity (*R*) derived from model simulations showing growth (vertical extension) of surviving ramets above the colony and erosion of dead ramets in 2018.



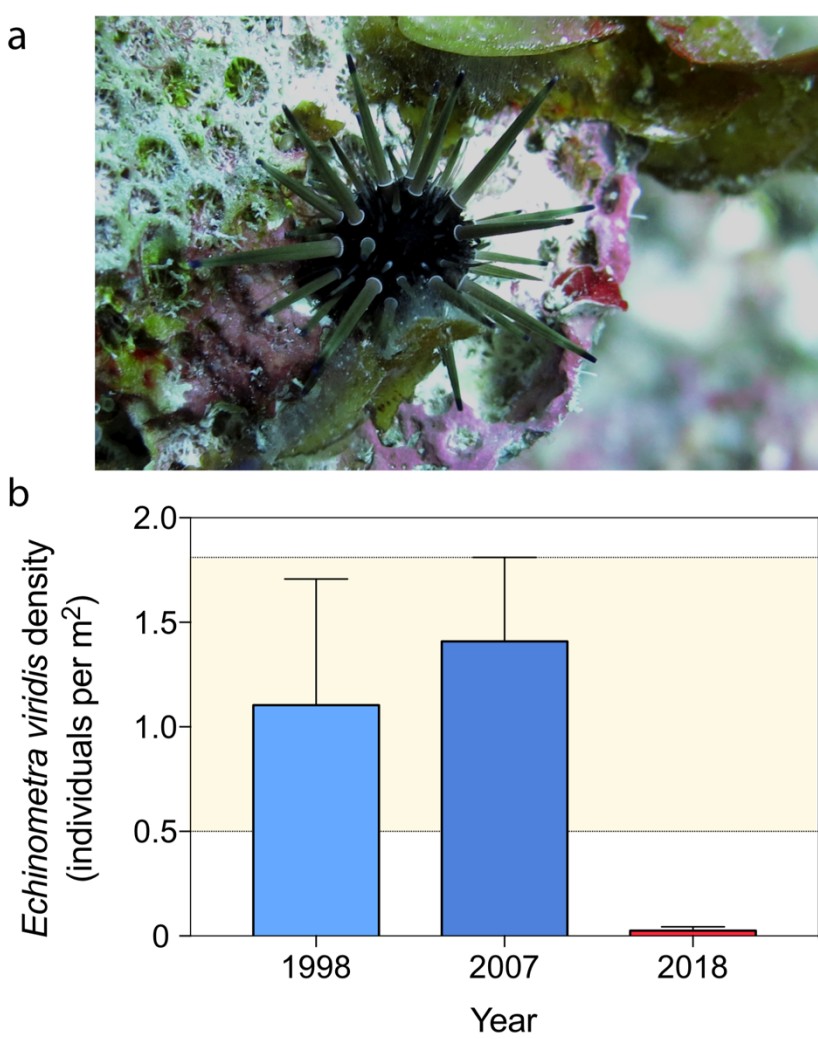


**Figure** 5 a) individual of *Echinometra viridis* among *Orbicella annularis* ramets at Long Cay in 2018 maintaining a cropped algal-free territory, b) density of *E. viridis* in 1998, 2007 and 2018 surveys (error band represents the maximum upper and lower standard deviation of the 1998 and 2008 surveys).
