# Peer review of "Multi-decadal changes in structural complexity following mass coral mortality on a Caribbean reef"

_Biogeosciences, 2019_

## Referee Comment (RC1) · Anonymous Referee #1 · 17 Oct 2019

This study combines field measurements of O. annularis survival/mortality, erosion, and linear extension across two decades following the 1998 bleaching event with a model of how those changes affect colony complexity over time. These sorts of fine-scale measurements of how coral mortality affect reef structural complexity and habitat partitioning are rare and it is, therefore, an important contribution to the literature. The manuscript is well-written and worthy of publication, but I think it could benefit from a more nuanced discussion of several aspects of the results, the most significant of which I've outlined below.

First, I would like to see the authors discuss how unique their results are to the specific

species of coral/habitat considered in the study. I suspect that the lack of change in colony-level rugosity is at least partially a result of the unique morphology and growth of O. annularis and the fact that the measurements were made continuously along the perimeter including the narrow gap between ramets (see my next comment). When most corals experience partial mortality (including other species of Orbicella), the majority of the regrowth by surviving ramets will be lateral, rather than vertical as in O. annularis. Therefore, the net positive growth of the corals in this study despite high levels of mortality, is not a result that would likely be observed in other species. The reef considered in this study is also fairly unique in its dominance by O. annularis. Although the authors are correct that Orbicella spp. have been dominant species on reefs throughout the Caribbean for more than a million years, many fore reefs have been dominated by O. faveolata rather than O. annularis, which is typically more common in lagoonal environments. While never stated outright, the manuscript implies from the very first sentence of the introduction that the reef considered in this study is typical of Caribbean reefs, but I would argue that the trajectories of erosion and complexity are likely very different elsewhere. I would like to see the authors more directly discuss how their observations from O. annularis at Long Cay may differ from what occurs for other species in other locations.

I would also like to see the authors put their rugosity estimates in the context of more traditional rugosity measurements that have been done on reefs and to discuss more explicitly what the observed changes in colony growth mean for the geometry of habitats within the colony. First, the rugosity estimates in this study are based on a theoretical continuous perimeter along the colony surface, whereas more traditional, chain-based methods of estimating rugosity vary based on the interval (chain link size) over which the measurements are made. In theory, the sort of continuous measurement estimated with the model in this study is a more accurate representation of overall rugosity, but from a practical standpoint it also gives a relatively high weight to the very narrow microhabitats between ramets. A traditional chain-based rugosity survey over the tops of these colonies would likely miss these microhabitats (which only have

an average opening of 0.4 cm) and would put more weight on coarse-level rugosity, which is likely low on living O. annularis colonies. This is important because while this study did not find significant colony-level rugosity changes over time, with the continued divergence between live and dead ramets, there likely were significant changes (increases) in more coarse-level rugosity that would be picked up with a chain-based measurement. The scale over which rugosity is considered is critical when thinking about how habitats may be changing over time. I think this is what the authors may have been suggesting in relation to the changes in the urchin populations, but changes in the sizes of microhabitats is not considered/discussed explicitly by the authors. It may be beyond the scope of this study, but one way to try tease apart changes in habitat over different scales would be to use the model presented in this study to look at the rugosity changes with varying theoretical measurement intervals ("chain link sizes"). A more minor, but related point is that while the authors suggest that the colony-level measurements represent "reef-scale" rugosity in several places in the manuscript, to my understanding they are only based on the height of the ramets, rather than on total colony elevation from the reef surface. That information and information about colony spacing would be needed to accurately estimate reef-scale rugosity.

I also think that it is important for the authors emphasize early in the Results and Discussion section that only vertical erosion was quantified/considered in this study. Although it is briefly discussed towards the end of the manuscript, the modeled changes in complexity do not consider erosion on the sides of the ramets, which is significant based on the images of the colonies. This erosion could have a significant effect on the perimeter values used to estimate rugosity in the model. This should at least be discussed as an assumption/source of uncertainty in the model.

Finally, I have some concerns about the discussion of the changes in urchin populations, which seem somewhat in conflict with the major conclusion of the study: that there were no significant changes in colony-level complexity over time. The authors suggest in the discussion that the decline in Eucidaris populations was a result of "observed long-term changes in complexity among crevices" (Lines 155-157), but the is at odds with the conclusion that structural complexity was "remarkably stable" (Line 107) over the time period. Furthermore, they suggest that bioerosion caused reduced crevice depth (Line 162), but the increase in the height difference between live and dead ramets over time seems to suggest just the opposite. The increased aperture of the openings between ramets because of erosion on their sides seems to be the most likely reason for increased predator access, but as mentioned previously, these changes were not measured/considered explicitly in this study. I don't necessarily disagree with the conclusion that urchin populations decreased because of increased access by predators after the coral mortality event, but I don't see how this conclusion is supported by the data they present.

Specific comments:

Line 10: "Th" should be capitalized throughout

Lines 22-23: I don't think this is true on many (most?) reefs anymore and this statement is not directly supported by the study cited at the end of the sentence. Although Orbicella spp. were historically the most abundant coral in Caribbean fore-reef environments, its abundance has declined significantly in many locations and the relative abundance of other taxa is now higher (as highlighted in Alvarez-Filip's studies for the Mesoamerican reef, specifically). I would re-word this sentence. It might be worth mentioning that this is a species complex not just O. annularis. Are the corals in this study O. annularis specifically? It looks like it from Fig. 1, but it would be good to make that clear in the methods.

Line 40: add a hyphen after "micro"

Lines 70-71: I think it might be helpful to add a sentence describing how these complexity measures are different from more typical, transect-level complexity measurements. Before digging into the code, it wasn't clear to me, for example, that the colony-scale complexity estimates were only measured for the top surface of the colony (right?), not

from its base.

Line 81/Figure 2c: The graph is labeled 2008, but the surveys were done in 2007, correct?

Line 82-84: This data should be summarized (means +/- SD) even if they were ns

Line 94: I would suggest changing "results in" to "resulted in" or perhaps just "drove"

Line 135-137: The colonies in the Keys were also 100% dead for the entire study period, so there was no potential for accretion. The two studies were also looking at different species of Orbicella, which have very different morphologies. Had there been surviving fragments of the O. faveolata colonies in the Keys, they would have likely expanded laterally before resuming any significant vertical growth.

Line 169-170: Is Long Caye a marine protected area? What is known about how invertivore populations have changed there over time?

Line 187: You are not looking at reef-scale complexity because your measurements are restricted to the top surfaces of the colonies. More broad-scale complexity relative to the seafloor is not considered.

Line 199: And 2003, correct? There are field photos from that year.

Line 204: A minimum distance?

Line 206: "it" should be "them", correct?

Line 210: were there also random factors included in the model?

Line 215: Where were the heights measured from? Not the base of the colony based on the values in the code. I'm guessing that it is the "height (i.e., vertical depth) of the ramet that parrotfish can graze, based upon field measurements." From Roff et al. 2015, but this is not clear in the text.

Line 235: O. annularis should be italicized Figure 1: Was there no bleaching in Belize

after 2010? What about disease?

Line 37 in the R code: The comment says minimum colony height was set at 2, but the value is 2.5. Thank you for providing your code!

---

## Referee Comment (RC2) · Brett Taylor (Referee) · 30 Mar 2020

**Brett Taylor (Referee)**

b.taylor@aims.gov.au

Received and published: 30 March 2020

The manuscript by Roff, Joseph, and Mumby uses a variety of techniques to determine long term changes in structural complexity, including microhabitat complexity, in a Caribbean reef framework based on growth and erosion of the major framework building coral Orbicella annularis. Their principal finding was that reef-scale structural complexity was relatively stable over time, despite extensive mortality of corals during the 1998 mass bleaching event, driven by rapid growth of surviving coral ramets. However, they found that microhabitat complexity declined substantially, and that this may have a considerably negative influence on cryptic fauna. As a representative case study, they

measured abundance the urchin Echinometra viridis over time, and this species nearly disappeared over the timeframe of this study.

Overall, I found this manuscript to be very well written with the methodology easy to follow. The rationale for the study was well justified and the results are compellingly robust and well interpreted. On these grounds I would recommend acceptance after minor revision. I see two areas where revisions may improve the quality of the interpretations:

1) Echinometra decline: the hypothesis of decline is compelling and supported by the observations. I feel it would improve the manuscript however to include any alternative hypotheses (if the authors can think of any) that might explain the decrease.

2) Only one coral species was studied (albeit importantly the major reef building species), but how well do the authors think the general results reflect patterns playing out in other major reef building corals, such as those growing laterally rather than vertically?

BGD

---

## Author Comment (AC1) · 15 May 2020

This study combines field measurements of O. annularis survival/mortality, erosion, and linear extension across two decades following the 1998 bleaching event with a model of how those changes affect colony complexity over time. These sorts of fine-scale measurements of how coral mortality affect reef structural complexity and habitat partitioning are rare and it is, therefore, an important contribution to the literature. The manuscript is well-written and worthy of publication, but I think it could benefit from a more nuanced discussion of several aspects of the results, the most significant of which I've outlined below.

[Figure]

Response: We appreciate the reviewer's thoughtful and detailed comments which have substantially improved the manuscript. We have included all revisions as suggested, and have outlined our responses and amendments below:

====

First, I would like to see the authors discuss how unique their results are to the specific species of coral/habitat considered in the study. I suspect that the lack of change in colony-level rugosity is at least partially a result of the unique morphology and growth of O. annularis and the fact that the measurements were made continuously along the perimeter including the narrow gap between ramets (see my next comment). When most corals experience partial mortality (including other species of Orbicella), the majority of the regrowth by surviving ramets will be lateral, rather than vertical as in O. annularis. Therefore, the net positive growth of the corals in this study despite high levels of mortality, is not a result that would likely be observed in other species. The reef considered in this study is also fairly unique in its dominance by O. annularis. Although the authors are correct that Orbicella spp. have been dominant species on reefs throughout the Caribbean for more than a million years, many fore reefs have been dominated by O. faveolata rather than O. annularis, which is typically more common in lagoonal environments. While never stated outright, the manuscript implies from the very first sentence of the introduction that the reef considered in this study is typical of Caribbean reefs, but I would argue that the trajectories of erosion and complexity are likely very different elsewhere. I would like to see the authors more directly discuss how their observations from O. annularis at Long Cay may differ from what occurs for other species in other locations.

Done. We appreciate the reviewer's point. First, we have amended the methods to clarify the monospecific stands of O. annularis in the present study and outline the O. annularis species complex to avoid confusion. Additionally we have included a broader description of the O.annularis species complex to highlight ecological differentiation among taxa within the complex:

"The study was conducted in Long Cay (Glovers Reef, Belize, Figure 1a). The reef framework at Long Cay is formed primarily from monospecific stands of Orbicella annularis (Ellis and Solander, 1786), which experienced widespread mortality following anomalously high water temperatures (29–32 °C) between early September and mid-November 1998 and hurricane Mitch which occurred simultaneously (Mumby, 1999). Field data were collected in 1998, 2003, 2007 and 2018 from an area of monospecific O. annularis dominated framework of approximately 400 m2 at a depth of 6-12m. O. annularis forms part of a species complex (the "Orbicella annularis species complex") along with O. faveolata and O. franksi. Each species within the complex exhibits a preferred depth zone, with O. faveolata dominating shallow reef habitats, O. annularis mid-depth habitats, and O. franksi in deeper depths (Pandolfi and Budd, 2008). (Lines 229-236)

Secondly, we appreciate the reviewer's point that trajectories of erosion and complexity will likely differ in other locations and among other closely related taxa. To highlight this point we have amended the results and discussion section to discuss the uniqueness of O. annularis frameworks more explicitly:

"High levels of genotypic diversity in O. annularis at Long Caye (Foster et al., 2013) and population connectivity to other reefs throughout the western Caribbean (Foster et al., 2012) implies that Long Caye is not unique, and differential growth of surviving ramets may lead to similar changes in structural complexity for O. annularis dominated frameworks elsewhere in the Caribbean (e.g. Idjadi and Edmunds, 2006; Edmunds and Elahi, 2007) where growth rates exceed erosion. At colony scales, changes in microhabitat complexity do not appear to have translated into changes in reef complexity, as the erosion of dead ramets is offset by growth of surviving ramets. This apparent stability in reef complexity at Long Caye is intrinsically linked to the columnar growth form of O. annularis colonies (Figure 2), and trajectories of erosion and structural complexity will likely vary among other Caribbean coral species with different morphologies (e.g. O. faveolata)". (Lines 140-147)

====

I would also like to see the authors put their rugosity estimates in the context of more traditional rugosity measurements that have been done on reefs and to discuss more explicitly what the observed changes in colony growth mean for the geometry of habitats within the colony. First, the rugosity estimates in this study are based on a theoretical continuous perimeter along the colony surface, whereas more traditional, chain-based methods of estimating rugosity vary based on the interval (chain link size) over which the measurements are made. In theory, the sort of continuous measurement estimated with the model in this study is a more accurate representation of overall rugosity, but from a practical standpoint it also gives a relatively high weight to the very narrow microhabitats between ramets. A traditional chain-based rugosity survey over the tops of these colonies would likely miss these microhabitats (which only have an average opening of 0.4 cm) and would put more weight on coarse-level rugosity, which is likely low on living O. annularis colonies. This is important because while this study did not find significant colony-level rugosity changes over time, with the continued divergence between live and dead ramets, there likely were significant changes (increases) in more coarse-level rugosity that would be picked up with a chain-based measurement. The scale over which rugosity is considered is critical when thinking about how habitats may be changing over time. I think this is what the authors may have been suggesting in relation to the changes in the urchin populations, but changes in the sizes of microhabitats is not considered/discussed explicitly by the authors. It may be beyond the scope of this study, but one way to try tease apart changes in habitat over different scales would be to use the model presented in this study to look at the rugosity changes with varying theoretical measurement intervals ("chain link sizes").

Response: We appreciate the reviewer's concern regarding the over-estimation of rugosity. The reviewer states that "A traditional chain-based rugosity survey over the tops of these colonies would likely miss these microhabitats (which only have an average opening of 0.4 cm)", but we highlight here that 0.4cm represents the minimum ramet

spacing, not the average ramet spacing. To determine ramet spacing, we measured 50 colonies of Orbicella annularis at Long Cay in 1998. The spacing between ramets was 4.7cm (see the nSpacing parameter in the supplementary model). A typical chain link of 0.7cm (e.g. Alvarez-Filip et al 2011) or nylon line as previously used in measurements of rugosity at Glovers Atoll (McClanahan 1999) would sufficiently capture such small-scale microhabitat complexity.

====

A more minor, but related point is that while the authors suggest that the colony-level measurements represent "reef-scale" rugosity in several places in the manuscript, to my understanding they are only based on the height of the ramets, rather than on total colony elevation from the reef surface. That information and information about colony spacing would be needed to accurately estimate reef-scale rugosity.

Done. We appreciate the reviewers point and have amended "reef-scale" to "colony-scale" throughout the manuscript for consistency.

====

I also think that it is important for the authors emphasize early in the Results and Discussion section that only vertical erosion was quantified/considered in this study. Although it is briefly discussed towards the end of the manuscript, the modeled changes in complexity do not consider erosion on the sides of the ramets, which is significant based on the images of the colonies.

Done. To emphasize the reliance on vertical erosion rates in the model we have included the following sentence in the first paragraph of the Results and Discussion:

"Prior to the 1998 mortality event, ramet heights within colonies were consistent, resulting in an even hemispherical colony appearance (Figure 1b). Surveys in 2018 revealed that surviving colonies exhibited a characteristic "serrated" topography, in that the isolated surviving ramets protruded above the remaining dead colony (Figure 2a, Figure

S1). As grazing parrotfish exhibit a strong tendency to erode the surfaces rather than sides of ramets (Roff et al., 2015), we focused on vertical erosion on the upper surfaces of dead ramets. We hypothesised that differences in growth among surviving ramets and the erosion of dead ramets would result in changes to microhabitat complexity" (Lines 85-90)

====

This erosion could have a significant effect on the perimeter values used to estimate rugosity in the model. This should at least be discussed as an assumption/source of uncertainty in the model.

Done. We appreciate the reviewers point. While our observations indicate that parrotfish (particularly Scarus) prefer to graze on upper surfaces over the sides of dead colonies (Roff et al 2015), bioerosion on the sides of ramets would likely weaken ramets and threaten structural integrity over long periods of time. To highlight this point we have included the following sentence in the discussion:

"While not explicitly incorporated in our erosion model, slower rates of external bioerosion on the sides of ramets and ongoing bioerosion from micro and macroborers over decadal scales (Roff et al., 2015) will likely weaken skeletal structural integrity in O. annularis (Highsmith et al., 1983), facilitating mechanical breakage and storm-driven loss of now protruding surviving ramets (Figure S1), likely resulting in non-linear increases of framework loss over longer timescales (2050 and beyond)" (lines 223-227)

====

Finally, I have some concerns about the discussion of the changes in urchin populations, which seem somewhat in conflict with the major conclusion of the study: that there were no significant changes in colony-level complexity over time. The authors suggest in the discussion that the decline in Eucidaris populations was a result of "long-term changes in complexity among crevices" (Lines 155-157), but the is at odds with

the conclusion that structural complexity was "remarkably stable" (Line 107) over the time period. Furthermore, they suggest that bioerosion caused reduced crevice depth (Line 162), but the increase in the height difference between live and dead ramets over time seems to suggest just the opposite. The increased aperture of the openings between ramets because of erosion on their sides seems to be the most likely reason for increased predator access, but as mentioned previously, these changes were not measured/considered explicitly in this study. I don't necessarily disagree with the conclusion that urchin populations decreased because of increased access by predators after the coral mortality event, but I don't see how this conclusion is supported by the data they present.

Done. We appreciate the reviewer's point. We note that the "long-term changes in microhabitat complexity among crevices" (Lines 155-157) is in agreement with the conclusion that structural complexity was "remarkably stable" (Line 107). We agree with the reviewer that increased aperture of openings is likely to have facilitated access by predators, and have expanded the discussion to include preferential feeding of parrotfish on the edges of dead coral substrates and widening of crevice apertures as follows:

"While further experimental work is needed to quantify size thresholds of refugia by which urchins escape predation, these observations are consistent with the hypothesis that bioerosion of reef frameworks results in reduced crevice depth for refuge, which in turn affects E. viridis densities by allowing for increased access for invertivorous fish, resulting in higher urchin mortality. Declines in the minimum ramet depth from 6.6 $\pm$ 3.9 cm in 2007 to 4.8 $\pm$ 2.1 cm (Figure 2c) are consistent with our previous U-Th estimates of bioerosion at Long Cay (Roff et al 2015). We hypothesise that as O. annularis ramets erode, the loss of the upper ramet lobes results in increased ramet spacing and a wider aperture of the crevices. As parrotfish preferentially target convex surfaces of dead coral substrates (Roff et al 2015), bioerosion of ramet edges can further widen the aperture of crevices, further facilitating access to invertivores and

[Figure]

diminishing refuge potential" (line 182-189)

====

Specific comments:

Line 10: "Th" should be capitalized throughout

Done.

====

Lines 22-23: I don't think this is true on many (most?) reefs anymore and this statement is not directly supported by the study cited at the end of the sentence. Although Orbicella spp. were historically the most abundant coral in Caribbean fore-reef environments, its abundance has declined significantly in many locations and the relative abundance of other taxa is now higher (as highlighted in Alvarez-Filip's studies for the Mesoamerican reef, specifically). I would re-word this sentence.

Done. We appreciate the reviewer's point and have reworded the sentence as follows:

"As an ecosystem engineer, Orbicella.annularis (Ellis and Solander, 1786) plays a critical ecosystem role as a framework building coral in the Caribbean (Geister, 1977) providing reef-scale structural complexity that supports a diverse range of fish (Alvarez-Filip et al., 2011) and invertebrate (Idjadi and Edmunds, 2006) assemblages" (Lines 22-24)

====

It might be worth mentioning that this is a species complex not just O. annularis. Are the corals in this study O. annularis specifically? It looks like it from Fig. 1, but it would be good to make that clear in the methods.

Done. We have included the following ecological description of the O. annularis species complex in the methods:

"O. annularis forms part of a species complex (the "Orbicella annularis species complex") along with O. faveolata and O. franksi. Each species within the complex exhibits a preferred depth zone, with O. faveolata dominating shallow reef habitats, O. annularis mid-depth habitats, and O. franksi in deeper depths (Pandolfi and Budd, 2008)" (Lines 234-236)

And have included mention of monospecific stands of annularis in the present study to avoid confusion:

"The reef framework at Long Cay is formed primarily from monospecific stands of Orbicella annularis (Ellis and Solander, 1786), which experienced widespread mortality following anomalously high water temperatures (29–32 °C) between early September and mid-November 1998 and hurricane Mitch which occurred simultaneously (Mumby, 1999). Field data were collected in 1998, 2003, 2007 and 2018 from an area of monospecific O. annularis dominated framework of approximately 400 m2 at a depth of 6-12m" (Lines 229-234)

====

Line 40: add a hyphen after "micro"

Done.

====

Lines 70-71: I think it might be helpful to add a sentence describing how these complexity measures are different from more typical, transect-level complexity measurements. Before digging into the code, it wasn't clear to me, for example, that the colony-scale complexity estimates were only measured for the top surface of the colony (right?), not from its base.

Done. We have added a sentence in the introduction to clarify the differences between typical complexity measurements:

"To determine changes in O. annularis frameworks at different scales, we calculated two metrics of habitat complexity: i) microhabitat complexity at the scale of individual ramets (centimetres), and ii) structural complexity at the scale of whole colonies (metres). These metrics consider the upper surfaces of O. annularis colonies, and differ from traditional transect-chain measurements of reef rugosity that assess structural complexity across multiple colonies (e.g. Alvarez-Filip et al., 2011)." (Lines 77-80)

and included a sentence in the methods to highlight how colony-scale estimates were included:

"To assess changes in rugosity at a colony scale in the two decades following the mass mortality, we created a structural model of O. annularis colonies parameterised using field data collected at Long Cay (see Supplementary code). The surface structural complexity of O. annularis colonies were modelled using a simple cross-sectional topography of ramets and colony widths (Figure 4a, see Supplementary R code)." (Lines 253-255)

====

Line 81/Figure 2c: The graph is labeled 2008, but the surveys were done in 2007, correct?

Done. Corrected Figure 2c to 2007

====

Line 82-84: This data should be summarized (means +/- SD) even if they were ns

Done. Included means as follows:

"No significant difference in height (p>0.05) was observed between "live-live" or "dead-dead" ramet pairings in either 2007 (0.5 ïĆś 1.0 cm, 1.2 ïĆś 0.9 cm) or 2018 (0.1 ïĆś 1.2 cm, 0.6 ïĆś 1.4 cm), implying that processes of growth and/or erosion occur evenly among living and dead ramets (Figure S1)" (Lines 93-95)

====

Line 94: I would suggest changing "results in" to "resulted in" or perhaps just "drove"

Done. Changed to "drove"

====

Line 135-137: The colonies in the Keys were also 100% dead for the entire study period, so there was no potential for accretion. The two studies were also looking at different species of Orbicella, which have very different morphologies. Had there been surviving fragments of the O. faveolata colonies in the Keys, they would have likely expanded laterally before resuming any significant vertical growth.

Done. Agreed - we have specified that the framework in the Keys study was dead O. faveolata:

"Long-term records of bioerosion over ecologically meaningful timescales are rare, yet a recent study (Kuffner et al., 2019) reporting exceptionally rapid rates of erosion of dead O. faveolata reef frameworks (maximum 1.63 cm yr-1) in the Florida Keys" (Lines 152-153)

====

Line 169-170: Is Long Caye a marine protected area? What is known about how invertivore populations have changed there over time?

Done. Long Caye was designated a marine reserve in 1993 and enforced since 1996. Invertivores were never heavily exploited prior to the reserve, and our survey data indicate no change in invertivore populations through time. We have expanded this section to include further discussion as follows:

"Higher biomass of invertivores inside of marine protected areas can substantially increase predation pressure on urchins (Harborne et al., 2009), and may explain the rapid decline in E. viridis at Long Caye following diminished refuge potential between

surveys. As Long Cay has been an enforced marine reserve since 1996, an alternative explanation to our observed data could be that urchin numbers have declined in response to increased predation pressure following recovery of invertivore fish assemblages. While plausible, we discount this hypothesis as invertivores were not heavily exploited prior to 1996 when the reserve was established, and surveys of fish assemblages indicate no change in invertivores over time (Mumby pers.obs.)." (Lines 190-196)

====

Line 187: You are not looking at reef-scale complexity because your measurements are restricted to the top surfaces of the colonies. More broad-scale complexity relative to the seafloor is not considered.

Done – amended to "colony scales"

====

Line 199: And 2003, correct? There are field photos from that year.

Done - corrected

====

Line 204: A minimum distance?

Done – corrected to "minimum distance"

====

Line 206: "it" should be "them", correct?

Done - corrected

====

Line 210: were there also random factors included in the model?

Done. Included "and "colony" as a random factor" at line 250

====

Line 215: Where were the heights measured from? Not the base of the colony based on the values in the code. I'm guessing that it is the "height (i.e., vertical depth) of the ramet that parrotfish can graze, based upon field measurements." From Roff et al. 2015, but this is not clear in the text.

Done. Ramet heights were measured from within colonies (i.e. the top to base of ramets). We have amended this as follows in the text:

"O. annularis colonies were modelled using a simple cross-sectional topography of ramets (Figure 4a, see Supplementary R code). Colony widths were determined from in-situ measurements of 95 colonies at Long Cay in 2000, and ramet heights (from the top to the base of the ramet within colonies) and widths measured from 30 ramets within colonies in 2000"

====

Line 235: O. annularis should be italicized Figure 1: Was there no bleaching in Belize after 2010? What about disease?

Done. As far as we are aware there are no reports bleaching or disease at Long Cay between 2010 and our survey in 2018, although minor bleaching was observed in 2019 after our study.

====

Line 37 in the R code: The comment says minimum colony height was set at 2, but the value is 2.5. Thank you for providing your code!

Done – corrected!

---

## Author Comment (AC2) · 15 May 2020

Overall, I found this manuscript to be very well written with the methodology easy to follow. The rationale for the study was well justified and the results are compellingly robust and well interpreted. On these grounds I would recommend acceptance after minor revision. I see two areas where revisions may improve the quality of the interpretations:

Response: We thank the reviewer for their careful and detailed comments.

====

1) Echinometra decline: the hypothesis of decline is compelling and supported by the observations. I feel it would improve the manuscript however to include any alternative hypotheses (if the authors can think of any) that might explain the decrease.

Response: As Long Caye was designated a marine reserve in 1993 and enforced since 1996, a plausible hypothesis would be that recovery in invertivore densities following enforcement may have placed increase predation pressure on urchins. However, invertivores were never heavily exploited prior to the reserve, and our survey data indicate no change in invertivore populations through time. We have expanded this section to include further discussion as follows:

"Higher biomass of invertivores inside of marine protected areas can substantially increase predation pressure on urchins (Harborne et al., 2009), and may explain the rapid decline in E. viridis at Long Caye following diminished refuge potential between surveys. As Long Cay has been an enforced marine reserve since 1996 an alternative explanation to our observed data could be that urchin numbers have declined in response to increased predation pressure following recovery of invertivore fish assemblages. While plausible, we discount this hypothesis as invertivores were not heavily exploited prior to 1996 when the reserve was established, and surveys of fish assemblages indicate no change in invertivores over time (Mumby pers.obs.)." (Lines 190-196)

====

2) Only one coral species was studied (albeit importantly the major reef building species), but how well do the authors think the general results reflect patterns playing out in other major reef building corals, such as those growing laterally rather than vertically?

Response: Following comments from Review #1, amended the methods to clarify the monospecific stands of O. annularis in the present study and outline the O. annularis species complex to avoid confusion.

The study was conducted in Long Cay (Glovers Reef, Belize, Figure 1a). The reef framework at Long Cay is formed primarily from monospecific stands of Orbicella annularis (Ellis and Solander, 1786), which experienced widespread mortality following anomalously high water temperatures (29–32 °C) between early September and mid-November 1998 and hurricane Mitch which occurred simultaneously (Mumby, 1999). Field data were collected in 1998, 2003, 2007 and 2018 from an area of monospecific O. annularis dominated framework of approximately 400 m2 at a depth of 6-12m. O. annularis forms part of a species complex (the "Orbicella annularis species complex") along with O. faveolata and O. franksi. Each species within the complex exhibits a preferred depth zone, with O. faveolata dominating shallow reef habitats, O. annularis mid-depth habitats, and O. franksi in deeper depths (Pandolfi and Budd, 2008).

Secondly, we appreciate the reviewer's point that trajectories of erosion and complexity will likely differ in other locations and among other closely related taxa. To highlight this point we have amended the results and discussion section to discuss the uniqueness of O. annularis frameworks more explicitly:

"High levels of genotypic diversity in O. annularis at Long Caye (Foster et al., 2013) and population connectivity to other reefs throughout the western Caribbean (Foster et al., 2012) implies that Long Caye is not unique, and differential growth of surviving ramets may lead to similar changes in structural complexity for O. annularis dominated frameworks elsewhere in the Caribbean (e.g. Idjadi and Edmunds, 2006; Edmunds and Elahi, 2007) where growth rates exceed erosion. At colony scales, changes in microhabitat complexity do not appear to have translated into changes in reef complexity, as the erosion of dead ramets is offset by growth of surviving ramets. This apparent stability in reef complexity at Long Caye is intrinsically linked to the columnar growth form of O. annularis colonies (Figure 2), and trajectories of erosion and structural complexity will likely vary among other Caribbean coral species with different morphologies (e.g. O. faveolata)".

---

## Author Response (AR2)

We sincerely thank both reviewers for their constructive and supportive comments during the peer review process. We have amended the manuscript to address the final comments from Reviewer 2 as follows:

L22: add a space between Orbicella annularis and delete the period. I would also specify here that you're talking about the O. annularis complex (or just Orbicella spp.) because many of the papers referenced in this paragraph (and elsewhere in the Introduction) do not talk about O. annularis specifically. It would be better to move the reference to O. annularis sensu stricto to the first sentence of the Results where you are describing your study site. From my perspective, it is important that the authors make this clarification.

Done. Corrected to Orbicella spp. throughout the introduction.

L26: I'd delete "modern day" as Gischler reconstructed Holocene accretion.

Done. Corrected to "Holocene"

L79: Perhaps add something like "our study" after "we focused"

Done.

L142: 9.4 should be -9.4

Done.

L159: There should be a period not a comma after the Lessios reference

Done.